# A Study of the Mechanical Properties of Composite Materials with a Dammar-Based Hybrid Matrix and Two Types of Flax Fabric Reinforcement

**DOI:** 10.3390/polym12081649

**Published:** 2020-07-24

**Authors:** Dumitru Bolcu, Marius Marinel Stănescu

**Affiliations:** Department of Mechanics, University of Craiova, 165 Calea Bucureşti, 200620 Craiova, Romania; dbolcu@yahoo.com

**Keywords:** hybrid matrix, flax fabric, composite materials, mechanical properties, chemical structure

## Abstract

The need to protect the environment has generated, in the past decade, a competition at the producers’ level to use, as much as possible, natural materials, which are biodegradable and compostable. This trend and the composite materials have undergone a spectacular development of the natural components. Starting from these tendencies we have made and studied from the point of view of mechanical and chemical properties composite materials with three types of hybrid matrix based on the Dammar natural hybrid resin and two types of reinforcers made of flax fabric. We have researched the mechanical properties of these composite materials based on their tensile strength and vibration behavior, respectively. We have determined the characteristic curves, elasticity modulus, tensile strength, elongation at break, specific frequency and damping factor. Using SEM (Scanning Electron Microscopy) analysis we have obtained images of the breaking area for each sample that underwent a tensile test and, by applying FTIR (Fourier Transform Infrared Spectroscopy) and EDS (Energy Dispersive Spectroscopy) analyzes, we have determined the spectrum bands and the chemical composition diagram of the samples taken from the hybrid resins used as a matrix for the composite materials under study. Based on the results, we have suggested that these composite materials could be used in different fields of activity.

## 1. Introduction

The composite materials based on renewable raw materials from agriculture and biomass are more and more used, because these products significantly compensate for the use of fossil fuels and reduce the greenhouse gas emissions in comparison with the petrol-based traditional materials. However, including natural fibers in polymers comes with several challenges, such as excessive absorption of water and low thermal properties, which must be overcome in order to manufacture materials with properties comparable with the traditional composite materials.

The synthetic resins have the disadvantage of a restriction of processing due to the high viscosity at melting, phenomenon which occurs at injection molding, and the final product is hard to be recycled. This disadvantage can be removed by using a vegetal oil resin-based thermorigid-biological matrix, which, being biodegradable, does not require a polymerization process (see [1,2,3]). Bio-polymers are obtained from renewable resources and, during recent years, they have acquired a higher importance compared with petrol-based polymers (see [4]). Among the most used vegetal resins are Sandarac, Copal and Dammar. These natural resins are insoluble in water, but are mildly soluble in oil, alcohol, turpentine and partially in petrol. With certain organic solvents, they form solutions usable as coverage polish (see [5,6,7,8,9]). These polishes have been used since the Middle Ages by people with the purpose of embellishing and protecting paintings, musical instruments or furniture. A disadvantage for the natural varnishes is given by the fact that it cannot form viscous resins (see for example the works [5,6]). A solution to remove this disadvantage is to use hybrid resins obtained by combining more elements, out of which at least one is organic and at least one is synthetic. It must be mentioned that most tests to obtain such resins have taken place in the industry of polishes (see [5,6,7]). Hybrid resins represent an environment friendly alternative in comparison with synthetic resins.

Besides the hybrid resins, the use of natural reinforcing materials instead of synthetic fibers has developed very much with a view to manufacturing biodegradable composite materials (see [10,11]). In comparison with synthetic fibers, natural fibers are characterized by biodegradability and renewability. Among the most used natural fibers, the following can be mentioned: kenaf, hemp, flax, jute, henequen, pineapple leaves, sisal, wood, herbage, corn stem, coconut, etc. According to [1,12] the specific strength and specific moduli of some natural fibers are comparable with those of the glass fiber. Instead, the ultimate tensile strength, Young’s modulus, density are lower in comparison with those of the glass fiber. The mechanical features, such as density, fiber diameter, tensile strength, Young’s modulus, elongation at break, for natural fibers (flax, hemp, jute, kenaf, ramie, nettle, sisal, henequen, abaca, palm oil, cotton, coconut) and synthetic fibers (glass fiber, carbon, Kevlar) are found in [10,11,12,13,14].

Among these natural fibers, flax, due to its higher tensile properties, has been applied widely as reinforcement material for polymeric composites. Flax fibers have some advantages in comparison with glass fiber, because these are less dense, renewable, combustible and have a relatively low price. The composite materials reinforced with flax are considered next generation materials due to the structural applications in various industries and especially in the automotive industry (see [12,13,14,15,16,17,18,19,20,21]. Various other applications of composites reinforced with flax and their properties are registered by [22], and the composites reinforced with flax and hemp are presented by [23,24,25].

An important restriction to successfully using the flax fibers in sustainable composite applications is their high humidity absorption and the low dimensional stability. Moreover, the mechanical properties of flax fibers are also affected by manufacturing processes, such as retting, scutching, bleaching and spinning (see [26,27]. To remove these shortcomings, various surface treatments for fibers have been done, including the treatment with silane, benzoylation and peroxide, to improve mechanical performance. The effects of surface treatment for flax fibers with vinyl trimethoxy silane (VTMO) and anhydride maleic-polypropylene (MAPP) on the mechanical properties of flax composites/PP have been researched by [28]. MAPP treatment has been adequate to flax/PP composites from the point of view of tensile strength and impact. VTMO treatment has shown higher bending properties and less influence on the impact properties following the absorption of humidity. Remarkable differences in mechanical behavior were found between alkaline treated fiber composites and untreated fiber composites (see [29]). It was found that the fiber-matrix interface was only marginally influenced by alkaline treatment. The main difference between treated and untreated fibers was the presence, in the mass of untreated fibers, of agglomerations of fibers. The combined action of alkalization and mechanical stresses during melt mixing led to a uniform distribution of treated fibers that were more efficient in obtaining superior mechanical properties, in accordance with theoretical predictions.

Blending hybrid resins and natural fibers fabrics leads to obtaining some environment friendly composite materials. Composite materials reinforced with flax threads and flax fabrics, with a strengthened-soy matrix, have been prepared by [30] to compare the properties of tensile strength and bending. The composites reinforced with flax threads have had the highest tensile strength of 298 MPa and a bending strength of 117 MPa, while the composites of flax fabrics have had a breaking point of 62 MPa in the warp direction and 83 MPa in the weft direction. The renewable epoxy resin resulted from vegetal oil is a potential substitute of the basic petrol resin in the flax composites used in applications for vehicle construction. Such composites are studies in [31,32,33,34,35], whereas [32] focuses on the influence of the architecture of fabrics on the mechanical properties of these composites. The tannin resin, a natural phenolic resin, reinforced with flax fibers, has been used to obtain some bodywork and dashboard elements for super-light electrical vehicles (see [36,37,38]. In the paper [39] composites based on tannin, reinforced with non-woven flax fibers have been manufactured and their mechanical properties have been studied. At the same time, Ref. [38,40] analyze the basic flax/tannin composites with four configurations of non-woven flax fibers and various angles of fabric layout.

Dammar, which is a resin obtained from trees from Dipterocarpaceae family from India and Eastern Asia, occupies a special place among the natural resins used. The studies performed on this resin have especially taken into consideration its chemical structure and chemical properties, presented in-depth in [20,21]. The resin mostly contains tetracyclic Dammarane skeletons, as well as pentacyclic oleanane, ursane and hopane derivatives. Also, it contains a small amount of sesquiterpenoid (C15). Dammar has been found to have an alcohol soluble part (α-resene) and an insoluble part (β-resene). Dammar resin is known to be used as a component of certain drugs. In this respect, Ref. [41] has conducted a study on the continuous administration of the drug atenolol made of Dammar gum reticulated with biodegradable hydrogel composites based on polyacrylamide and zirconium. The reticulation process has been successfully synthesized, obtaining better properties for the polyacylamide Dammar gum, which are explained by the presence of an inorganic component in the zirconium mixture. The hydrogel properties (Dammar gum mixed with zironium iodoxalate) are analyzed by [42,43,44]. The samples were prepared to incorporate certain inorganic zirconium precipitates in a polymer mixture under vacuum conditions. The morphology and structure have been studied by using FTIR (Fourier transform infrared spectroscopy), SEM (Scanning Electron Microscopy) with EDS (Energy dispersive spectroscopy), X-ray diffraction data, Thermo gravimetric analysis, differential thermal analysis and differential thermo gravimetric analysis. In the end, we have found that hydrogel can be used for preferential removing of $P_{b}^{2-}$, in comparison with other metal ions, in solutions. Also, it has shown high chemical and thermal stability.

Studies concerning the mechanical behavior of natural resins are few. In [45], the mechanical features (tensile strength, percentage elongation and Young’s modulus), water vapor transmission features and humidity absorption features for films of Dammar which contain or do not contain plasticizer have been studied. Ref. [46,47] study the method by which the addition of Dammar has contributed to the improvement of rigidity, elasticity modulus and hardness of a modified silicon. Studies referring to composite materials which have matrix and reinforcing material of natural materials are few. In [48], the mechanical properties of some composite materials obtained of hybrid resin based on Dammar reinforced with flax, cotton, hemp, wheat straw and bulrush are studied. In [49], the influence of some non-uniformities on the mechanical behavior of composite materials with Dammar-based matrix and reinforcing material of hemp fabric is presented.

There are no significant results regarding the mechanical properties of Dammar natural resin. This is due to the shape of the granules in which the Dammar resin is found (a shape that does not allow the making of samples to be subjected to the tensile test).

In this paper, we study some mechanical properties of some composite materials, which have as matrix three types of Dammar-based hybrid resins, and, separately, for each type of resin, we have used as reinforcing material two types of flax fabric.

## 2. Used Materials and Equipment

### 2.1. Making the Samples

The temperature of the environment in which we made the castings was 21–23 °C. All specimens were cut 10 days after pouring the plates.

Please note that the natural Dammar resin we used to make the hybrid resin was purchased from an authorized distributor (VITALNET MED SRL, Bucharest, Romania, see [50]), and the two types of flax fabric were purchased directly from the manufacturer (DINATEX SRL, Falticeni, Suceava, Romania, see [51]).

The natural Dammar resin is diluted with turpentine and if it is kept in closed recipients, it is remains liquid. That mixture is used for the protection of paintings (varnishes). The disadvantage is that the resin strengthening process is very long, even if it is applied in thin layers. Ref. [5,6,7] show that natural varnishes can create thick resins only in the presence of some synthetic components. We have removed this shortcoming by adding synthetic resin, together with the associated hardener.

During the first phase, we have cast three plates of resin.

We have done as follows:we have mixed a volume ratio of 50%, 60% and respectively 70% of liquid Dammar with 50%, 40% and respectively 30% epoxy resin of Resoltech type 1050 and the associated hardener of Resoltech 1055 type;in the end, to increase the volume ratio of Dammar from the obtained resins (and to increase the viscosity and implicitly to reduce the hardening time), we have added 10% Dammar powder of the total volume and we have mixed until homogenization.

Henceforth, the three types of obtained resin will be called hybrid resins of type 1, 2 and 3.

The properties of the epoxy resin (Resoltech 1050/Resoltech 1055) can be accessed from the webpage of the manufacturer (see [52]). The synthetic structure (epoxy resin and hardener) has been necessary to generate fast activation points for the polymerization process.

We have cut three sets of ten samples from the cast plates and they will be labelled with: 1.1–10 for samples of hybrid resin of type 1; 2.1–10 for samples of hybrid resin of type 2; 3.1–10 for samples of hybrid resin of type 3. The cutting of samples has been done 10 days later than the plates casting. The samples sizes have been: length of 250 mm, width 25 mm and thicknesses have been 6.1 mm and 6.2 mm. These dimensions are used in the tensile tests, which are done according to ASTM D3039 (see [53]). The density of samples is between 1.04–1.06 g/cm${}^{3}$.

In Figure 1, a sample from each set of hybrid resin is presented.

During the second phase, we have manufactured composite materials by reinforcement with five layers of two types of flax fabric from the hybrid resins. The flax fibers properties are (see for example [1,54,55]): density 1.5 g/cm${}^{3}$, elasticity modulus 27–39 GPa, tensile strength 345–1100 MPa, elongation at break 2.7–3.2%. When making the composite plates we applied a uniform pressure of 27,000 N/m${}^{2}$.

For the first type, 12 layers of mixture with 20% cotton and 80% flax have been placed, the fabric with the specific mass of 240 g/m${}^{2}$. The obtained composites have a density of 1.17–1.18 g/cm${}^{3}$, and the mass ratios of resin in the manufactured plates are presented in Table 5.

For the second type, 20 layers of mixture with 40% cotton and 60% flax have been placed, the fabric with a specific mass of 160 g/m${}^{2}$. The obtained composites have a density of 1.20–1.21 g/cm${}^{3}$, and the mass ratios of resin in the manufactured plates are presented in Table 6.

The two types of flax fabric are presented in Figure 2.

We can see that in the composites reinforced with the same type of fabric, no significant differences of density exist when the Dammar ratio is modified in the used hybrid resin. The composites reinforced with the second type of fabric have a higher density than those reinforced with the first type of fabric, which can be explained by the differences of volume ratio and by the differences of density between fabrics.

We have also cut sets of ten samples from the plates of composite materials. The samples have had a length of 250 mm, width of 25 mm and thickness between 6.2 mm and 6.3 mm, for the samples reinforced with first type of fabric and between 6.1 mm and 6.2 mm for samples reinforced with second type of fabric. For the study of vibrations, samples with a length of 300 mm and width of 25 mm have been cut from plates of composite materials.

Table 1 presents centralized the constituent elements of the composite materials made.

### 2.2. Devices Used for Tests

Both the samples of hybrid resin and those of composite materials have been subject to tensile test, which has been done in accordance with the provisions of ASTM D3039. We have used the machine for mechanical tests LLOYD Instruments Lrx PLU, with maximum force of 2.5 kN, equipped with the analysis soft NEXYGEN (technical features can be found on the web page of the manufacturer, see [56]).

The traction speed was 2.5 mm/min.

The mechanical properties obtained based on this test have been: characteristic curve, tensile strength $R_{m}$ [MPa], percentage elongation at break *A* [%] and elasticity modulus *E* [MPa].

The FTIR analysis has been performed by a portable system of IdentifyIR type (equipped with the ChemAssist soft). It complied with ASTM E168 and E1252 (see [57,58]). The size of the used sample has been under 5 μL or 100 μm. Moreover, the equipment has been provided with ATR spectral libraries for detection Aldrich/Smiths. All the technical features of this system can be found on the web page of the manufacturer (see [59]).

The SEM-EDS analysis has been done with an electronic microscope Hitachi model S3400N-type N, equipped with EDX Oxford Instruments X-act, with traditional cathode (see [60] with the technical specifications given by the manufacturer). Moreover, we mention that for the EDS analysis ASTM E1508 was complied with (see [61]).

For the study of vibrations, the following measurement device has been used: data acquisition system SPIDER 8, connected by USB to a notebook; data acquisition set done by CATMAN EASY soft, which has connected the two entities; signal conditioner NEXUS 2692-A-0I4 connected to the SPIDER 8 system; accelerometer with sensitivity $0.04\times 10^{-12}$ C/ms${}^{-2}$, connected to signal conditioner.

## 3. Results

Furthermore, by representative sample of a set we will understand a sample with average values of the studied mechanical properties.

### 3.1. Experimental Results for the Studied Hybrid Resin

In Figure 3, the tensile test equipment for a sample of hybrid resin is presented.

The characteristic curves for a representative sample of each set of samples from the obtained hybrid resins are presented in the Figure 4.

The experimental results for sets of samples of hybrid resin are presented in Table 2.

We notice an important modification of the mechanical behavior, materialized by a change in the characteristic curve. If, in the case of 50% Dammar natural resin samples the characteristic curve is practically linear, in the case of the other types of samples we observe the occurrence of a nonlinearity area, specific plastic behavior. These nonlinearities appear during specific deformations higher than 1.5%, being more obvious in the case of 70% Dammar (which have the highest specific deformations). This phenomenon can be explained by the fact that the resins, even in solid state, have a rheological behavior characterized by viscosity. Due to external stresses, a low viscosity leads to significant deformations. Dammar resin diluted with turpentine remains liquid, so it has a low viscosity. Solidification occurs by mixing with synthetic resin. We found that the viscosity of the hybrid resin obtained decreases with increasing volume ratio of Dammar. Therefore, at the same external load, the deformations of the hybrid resin by 70% Dammar are higher than the deformations of the hybrid resin by 60%, respectively 50% volume ratio of Dammar. In-depth studies on the composition and chemical properties of the hybrid resin based on Dammar (with various volume ratio) were performed in the paper [62].

In Figure 5, we present the FTIR analysis of a specimen of hybrid resin (specimen from type 2) and of the same re-tested specimen. The infrared spectrum, characteristic bands and a list of peaks seen between 4000–650 cm${}^{-1}$ are presented. This spectrum diagram is automatically plotted by the analysis system software.

Based on the diagram (and the peak list) two functional groups can be seen:Ketene cumulated double bonds $\left(CH_{2}=C=O\right)$, which are substance with cumulated carbonyl and carbon–carbon double bonds;nitrogen double bond, known as Nitrite ion ($NO_{2}^{-}$) and which contains nitrogen in a relatively unstable oxidation condition.

The FTIR analysis equipment is equipped with Aldrich/Smiths ATR spectral libraries that can automatically identify a maximum of 10 volatile organic compounds in the structure of the analyzed resin. In Table 3, we present 10 volatile compounds identified in the chemical structure of hybrid resin of type 2.

Based on the EDS analysis of a specimen of hybrid resin of type 2, taken from the samples of 2.x type, in Figure 6 we show the diagram of the chemical composition obtained at an intensity of 15 keV.

In Table 4, based on the EDS analysis, we present the chemical structure of a specimen of hybrid resin of type 2, taken from sample 2.x. This structure is expressed by weight concentration, atomic concentration and number of atoms of each element.

We have used a total number of atoms equal with $48\times 10^{23}$ approximately.

As in the paper [48], after the analysis of chemical concentrations in the three types of hybrid resin, we can conclude that:-for Carbon, a decrease in the Atomic Concentration and the Weight Concentration is registered as the volume ratio of Dammar increases;-for Oxygen, an increase in the Atomic Concentration and the Weight Concentration is registered as the volume ratio of Dammar increases.

### 3.2. Experimental Results for the Studied Composite Materials

The samples manufactured from composite materials reinforced with flax fabric have been subject to the tensile test. Figure 7, Figure 8 and Figure 9 show the characteristic curves for composite materials with matrix of the three types of hybrid resin and which have been reinforced with the first type of flax fabric.

The experimental results for sets of samples from composite materials with matrix of three types of hybrid resin, reinforced with first type of flax fabric, are presented in Table 5.

Figure 10, Figure 11 and Figure 12 show the characteristic curves for composite materials with matrix of the three types of hybrid resin and which have been reinforced with the second type of flax fabric.

The experimental results for the sets of samples of composite materials with matrix of the three types of hybrid resin, reinforced with the second type of flax fabric, are presented in Table 6.

By comparison, composites made with the two types of flax fabrics have similar behaviors. However, composites reinforced with the first type of fabric have a breaking strength with 4.5–5 MPa higher than composites reinforced with the second type of fabric. In addition, for the modulus of elasticity the conclusion is similar. Composites reinforced with the first type of fabric have a modulus of elasticity with 285–450 MPa higher than composites reinforced with the second type of fabric. These differences arise from variations in the properties and distribution of flax fibers in the fabric, respectively.

In Figure 13, we present the image of the breaking area for a representative sample of the set of composite materials with matrix of hybrid resin of type 2 and reinforcing material of the first type of flax fabric (a), respectively the second type of flax fabric (b).

The studies on the vibrations of bars made of composite materials are based on models of various deformation theories that take into account shearing and are adapted so as to evaluate the static and dynamic characteristics of bars. Such a model is the Rayleigh model which considers the rotational inertia of the bar section. A more complex model is the Timoshenko model, which considers deformations due to shearing stress as well. This model underlies the “first order shear deformation theory”, symbolized by FSDT. All these models consider that a plane and normal section on the average fiber before deformation stays plane without keeping the specified perpendicularity. The limits of the elementary theory and FDST have required the introduction of higher order shear deformation theories, symbolized HSDT. The main differences between the suggested models are generated by the functions used for ascertaining the deviations from the plane character of the bar section submitted to (polynomial, trigonometric, deformation hyperbolic) deformation. [63] compares the results obtained with the help of this theory.

In reality, due to internal friction and interaction with the air, all vibrations are damped. A general study conducted by [64] presents three mechanisms of energy dissipation. The first introduces the so-called external or viscous damping; however, for this mechanism all the modal amplitudes are damped at the same ratio, contrary to experience. The second mechanism considers that the damping force is proportional to the bending speed, whereas the third mechanism considers that the damping ratios of the specific vibration modes depend proportionally on the square of the frequency. A good number of articles investigate the various aspects and mechanisms of this phenomenon and the influence on vibration behavior of different composite materials (see [65,66,67,68,69,70,71]).

We have experimentally determined the damping coefficient and natural frequency for samples of composite materials having as matrix the three types of hybrid resin and reinforced, separately, with the two types of flax fabric. The studied samples have been embedded at one head, and the measurement has been done at the free head. The free length of samples has been 150 mm, 175 mm, 200 mm, 225 mm and 250 mm.

In Figure 14, we present the record of vibrations (natural frequency and damping factor) in a sample of composite material with matrix of hybrid resin of type 2, reinforced with the first type of flax fabric, for a free length of 200 mm.

In Table 7, we present the behavior under vibrations of samples from materials for composites with matrix of hybrid resin of type 1, 2 and 3, reinforced with first type of flax fabric. The presented values represent the arithmetic mean for three measurements.

In Figure 15, we present the recording of vibrations (natural frequency and damping factor) in a sample of composite material with matrix of hybrid resin of type 2, reinforced with the second type of flax fabric, for a free length of 200 mm.

In Table 8, we present you the behavior at vibrations of samples from materials for composites with matrix of hybrid resin of type 1, 2 and 3, reinforced with the second type of flax fabric. The presented values represent the arithmetic mean for three measurements.

The natural frequencies of the bars depend on not only the dimensions (thickness, length), but also on the material properties (density and modulus of elasticity).

Because the test samples studied had similar dimensions, the differences between the measured frequencies are due to the differences between the modulus of elasticity of the composite materials from which the samples are made.

We found that there is a correspondence between the measured frequencies and the modulus of elasticity determined at the tensile test, more precisely the increase of the modulus of elasticity leads to the increase of the vibration frequency. The damping factor as a whole characterizes the vibration damping capacity for a test piece. Comparison of the results in Table 7 and Table 8 shows that the samples reinforced with the second type of fabric dampen the vibrations better. We also found that the vibration damping capacity increases with increasing volume ratio of Dammar in the hybrid resin used as a matrix.

## 4. Discussion

The use of natural resins for manufacturing composite materials is influenced by the properties of these resins and by their capacity of performing a synergetic effect together with the reinforcement materials. The analysis of the obtained results shows a significant variation of the hybrid resins properties studied depending on the ratio between the natural and synthetic resin. A quick decrease in the elasticity modulus is seen at the increase of the Dammar ratio in the structure of the hybrid resin, from 3077 (± 92) MPa for the hybrid resin of type 1, to 1798 (± 45) MPa for the hybrid resin of type 3. A decrease from 25.5 (± 1.5) MPa for hybrid resin of type 1, to 13 (± 1) MPa for hybrid resin of type 3 can also be noticed. For the elongation at break, the effect is reverse, an increase is seen once as the Dammar volume ratio increases. Significant modifications can be seen in the form of the characteristic curve too. If in the case of the hybrid resin of type 1 the characteristic curve is almost linear, in the hybrid resin of type 3 the nonlinearity is significant.

For studied composite materials the comparison of experimental results shows a significant modification of the mechanical properties when the ratio between the epoxy resin and natural resins is changed. A decrease in values for the tensile strength and the elasticity modulus is seen as the volume ratio of natural resin from the mixture increases. The elongation at break, although it increases in the case of hybrid resins as the Dammar ratio increases, in the case of composites, it decreases as the Dammar ratio increases, regardless of reinforcing materials.

The analysis of the characteristic curves shows that three stages of the stress process stand out. In the first stage, we notice a proportionality between the normal tension and the specific deformation, so Hooke’s law is applicable. Therefore, at this stage, we may consider that the stress is taken over uniformly by matrix and reinforcer. In the second stage, the composite material displays a nonlinear behavior, as the dependence between the normal tension and the specific deformation loses its proportionality character. In this stage, we may conclude the tensile tension reaches a maximum in the resin and, consequently, it may present breaks, tack losses between fibers and matrix, as well as wrenches of the fibers from the resin. Thus, there is a stress transfer to the fibers. If in the first stage and the beginning of the second one the stress is taken over by the whole composite material, at the end of the second stage the stress is mainly taken over by the fibers. In the third stage the dependence between the normal tension and the specific deformation becomes linear again, which can be explained by the stress being taken over by the fabric fibers only, which are longitudinally arranged, and the material break occurring when the fibers break up.

The behavior of composite materials is also confirmed by the images obtained using SEM analysis, which shows that the fibers, in the breaking area, are detached from the matrix. In addition, a higher fiber density is observed for the first type of flax fabric, which explains the superior properties of composites reinforced with this type of fabric.

The variations of tensile strengths, both for composites reinforced with the first type of fabric and for the composites reinforced with the second type of fabric, are smaller than the variations of tensile strengths for the used resins. If, for the hybrid resin, the maximum variations between the tensile strength of type 1 resin and the tensile strength of type 3 resin has been 15 MPa, in the composite materials, regardless of the reinforcing materials, this difference has been 9 MPa. In the case of the elasticity modulus of composite materials, these variations have been even smaller. If, for the hybrid resin, the maximum variation between the elasticity modulus of the hybrid resin of type 1 and elasticity modulus of hybrid resin of type 3 has been 1417 MPa, for the composites reinforced with the first type of fabric this variation has been 430 MPa, and for the composites reinforced with the second type of fabric the variation has been 330 MPa. This phenomenon can be explained by the fact that the stress has been taken from the beginning by fibers, the matrix having the role to support the fibers.

The properties of flax fibers are influenced not only by natural factors (climatic conditions, soil, seed varieties, etc.), but also by plant processing conditions. We can find, based on bibliographic data, that the mechanical properties of flax fibers (see [1,4]) are far superior to the mechanical properties of the hybrid resins studied by us. And in the case of studied composite materials, we found that the values of modulus of elasticity and those of tensile strength are much higher than those of hybrid resins used as matrices. Therefore, we can conclude that the two types of flax fabrics have a decisive influence in establishing their mechanical behavior. In addition, the elongation at break in composite materials reinforced with the two types of flax fabric is close to the elongation at break of flax fibers which is 2.7–3.2% (see [1,4]). Therefore, the breaking of the test pieces from the studied composite materials took place at the time of breaking the fibers.

Because the damping factor depends on the sample length, it characterizes the global capacity of the sample damping. In order to determine the capacity to damp vibrations in the studied composite materials, the loss factor can be calculated as $\eta =\frac{\mu }{\pi \nu }$ (see [48]) for each of the materials. For composite materials reinforced with the first type of flax fabric, the average value of the loss factor is:-η=0.0749 for composite with matrix of resin of type 1;-η=0.0804 for composite with matrix of resin of type 2;-η=0.0895 for composite with matrix of resin of type 3.

For composite materials reinforced with the second type of flax fabric, the average value of the loss factor is:-η=0.1129 for composite with matrix of resin of type 1;-η=0.1232 for composite with matrix of resin of type 2;-η=0.1363 for composite with matrix of resin of type 3.

The loss factor for composites reinforced with resin with type 3 is 20% higher than for the composites reinforced with resin of type 1. Consequently, we obtain an increase in the capacity to damp vibrations, when there is an increase of the Dammar ratio in the composition. For the composite materials reinforced with the second type of fabric, the loss factor is 50% higher than for the composites reinforced with fabric of first type. This can be explained by the higher number of layers, but it can also be due to the differences of properties of the two types of fabrics. We notice that the damping factor is inversely proportional with the square of the free length of the bar. Since there is a similar dependence in the case of the own pulsations we may draw the conclusion that it is predominantly a mechanism of energy dissipation, where the damping force is proportional to the bending speed of the bar.

For the elongation at break, even though it increases in the case of hybrid resins when there is an increase in the Dammar ratio, in the case of composites, regardless of the reinforcing material, it decreases with the increase in the Dammar ratio. This shows that the composite break up when the break of fibers takes place.

Based on the list of peaks seen on the spectrum of the hybrid resin of type 2, we can determine the following groups and classes of chemical components:-2916–2870 cm${}^{-1}$ the elongation of aromatic and aliphatic C−H bonds takes place (saturated systems of alkane type);-1450 cm${}^{-1}$ the elongation of aromatic C−H bonds takes place (indicates the presence of methylene groups $\left(CH_{2}\right)$);-1242 cm${}^{-1}$ the elongation of C−O−C bonds of ethers and esters takes place;-750 cm${}^{-1}$ indicates the presence of some strong bonds of C−C1 type.

Comparing the spectrums of the three types of hybrid resin, we notice a mitigation of the absorption capacity only; no shrinking or movements of bands are observed, which suggest the lack of interaction between components.

## 5. Conclusions

The paper studied the mechanical properties of composite materials made of three types of hybrid resin based on Dammar reinforced with two types of flax fabric. We found that

for hybrid resins the modulus of elasticity and breaking strength decrease, and the elongation at break increases with increasing volume ratio of Dammar;for composite materials both the modulus of elasticity, the breaking strength and the elongation at break decrease with increasing volume ratio of Dammar;the studied composite materials have good vibration damping properties; the loss factor increases with increasing volume ratio of Dammar;composites reinforced with the first type of flax fabric have better mechanical properties than composites reinforced with the second type of fabric, but have lower damping properties;for all studied composites the damping factor is inversely proportional to the square of the bar length; this corresponds to an energy loss mechanism in which the damping force is proportional to the bending speed of the bar.

The properties of the studied composite materials show that they can be used successfully in:
-aeronautics for manufacturing some fuselage elements or movable empennage elements, flaps, and so on;-construction of vehicles for manufacturing some bodywork elements, such as wings, doors, and so on;-civil and industrial engineering for manufacturing some component elements of formworks as substitute for composite materials manufactured of synthetic components.

## Figures and Tables

**Figure 1 polymers-12-01649-f001:**
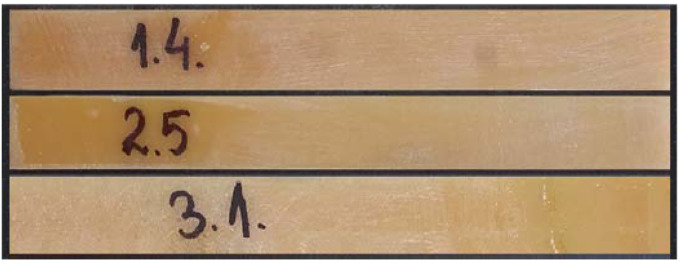
Samples of each set of hybrid resin.

**Figure 2 polymers-12-01649-f002:**
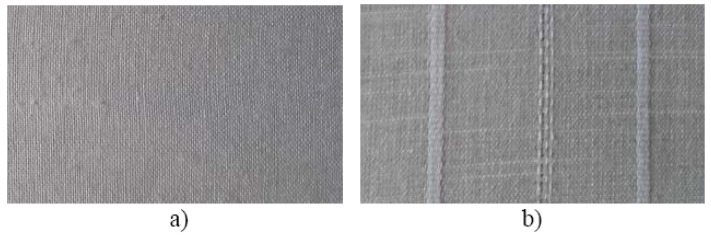
Used flax fabrics: (**a**) 20% cotton and 80% flax; (**b**) 40% cotton and 60% flax.

**Figure 3 polymers-12-01649-f003:**
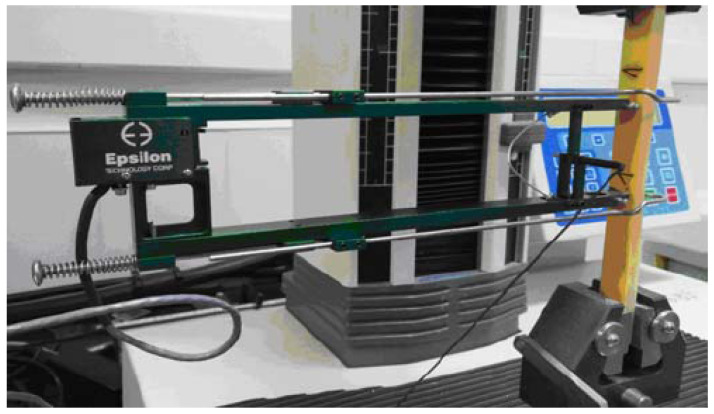
The tensile test equipment for a sample of hybrid resin.

**Figure 4 polymers-12-01649-f004:**
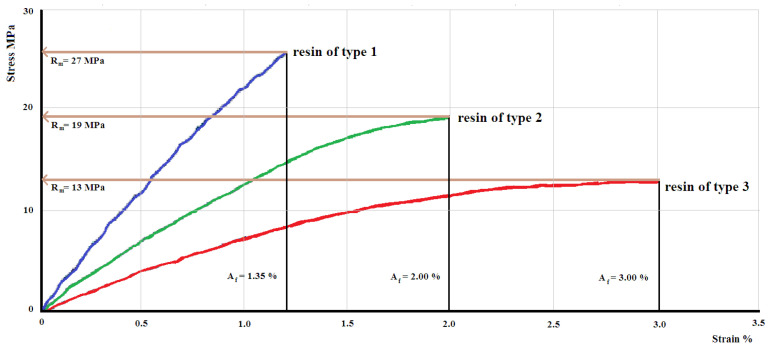
Characteristic curves for a representative sample of each set of samples from the obtained hybrid resins.

**Figure 5 polymers-12-01649-f005:**
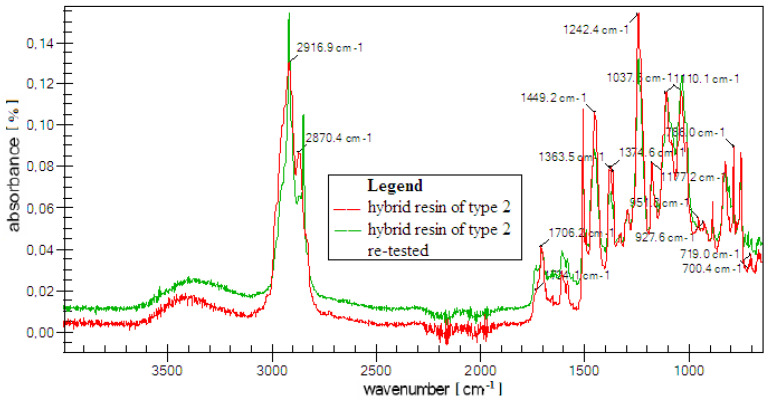
FTIR analysis of a specimen of hybrid resin (specimen from type 2) and of the same re-tested specimen.

**Figure 6 polymers-12-01649-f006:**
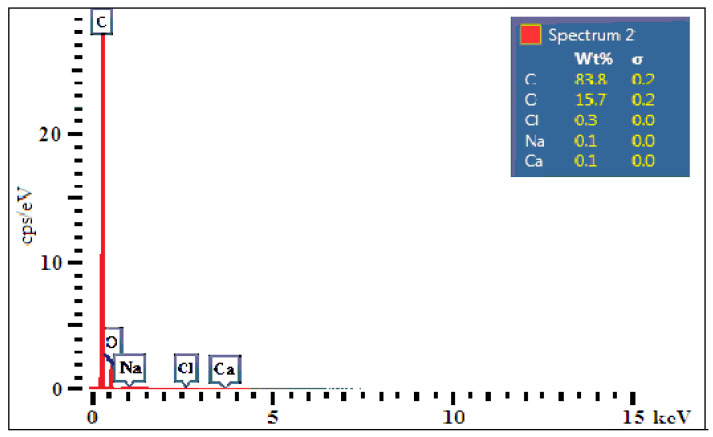
EDS analysis diagram of the chemical structure of a specimen of hybrid resin of type 2, taken from sample 2.x., obtained at an intensity of 15 keV.

**Figure 7 polymers-12-01649-f007:**
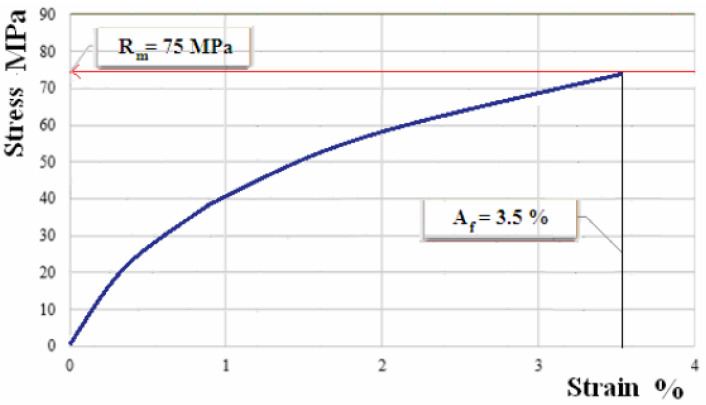
Characteristic curve for sample of composite material with matrix of hybrid resin of type 1, reinforced with fabric of first type.

**Figure 8 polymers-12-01649-f008:**
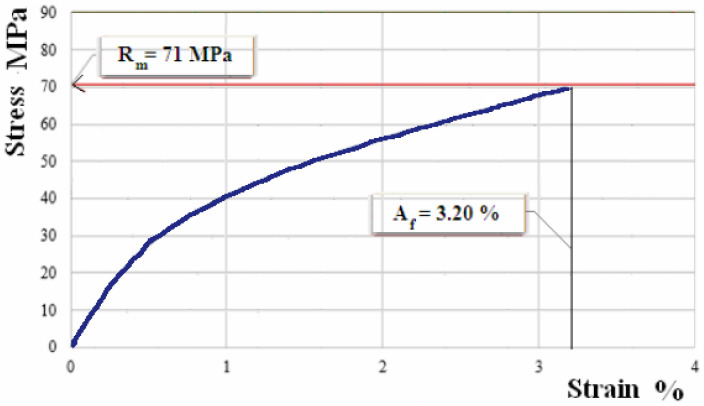
Characteristic curve for sample of composite material with matrix of hybrid resin of type 2, reinforced with fabric of first type.

**Figure 9 polymers-12-01649-f009:**
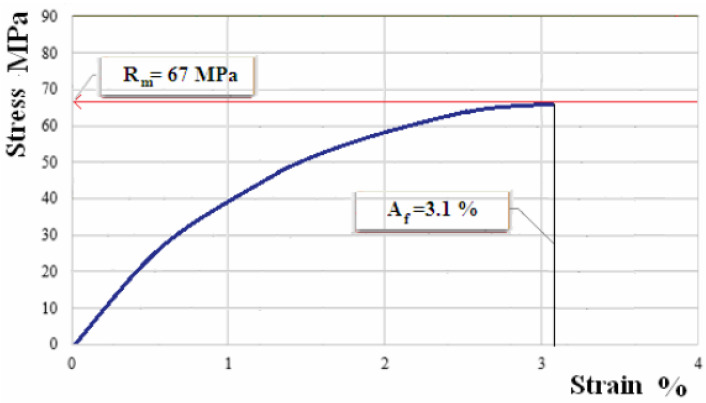
Characteristic curve for sample of composite material with matrix of hybrid resin of type 3, reinforced with fabric of first type.

**Figure 10 polymers-12-01649-f010:**
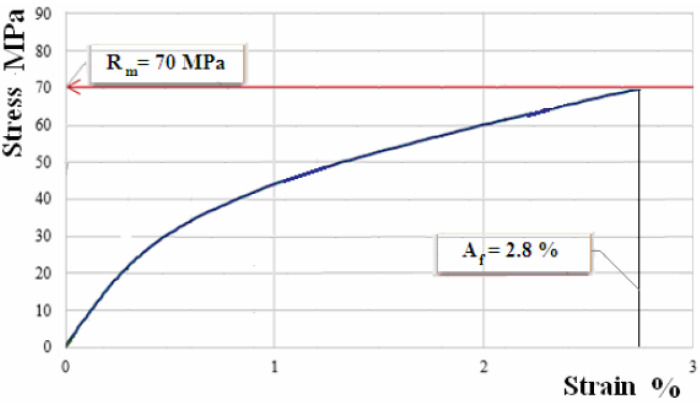
Characteristic curve for sample of composite material with matrix of hybrid resin of type 1, reinforced with the second type of fabric.

**Figure 11 polymers-12-01649-f011:**
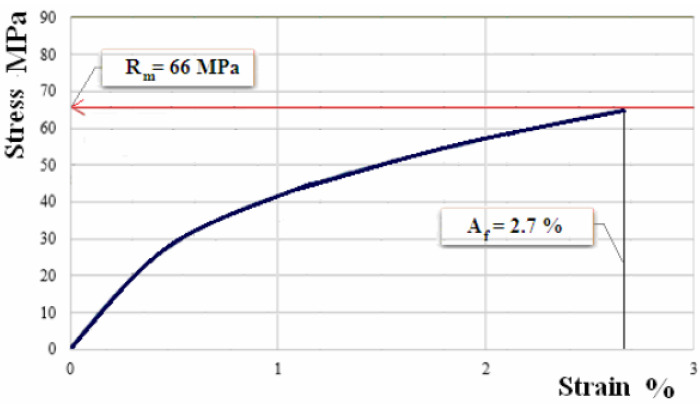
Characteristic curve for sample of composite material with matrix of hybrid resin of type 2, reinforced with the second type of fabric.

**Figure 12 polymers-12-01649-f012:**
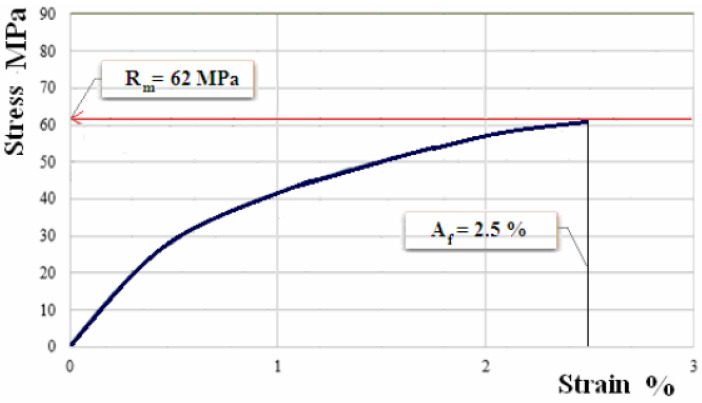
Characteristic curve for sample of composite material with matrix of hybrid resin of type 3, reinforced with the second type of fabric.

**Figure 13 polymers-12-01649-f013:**
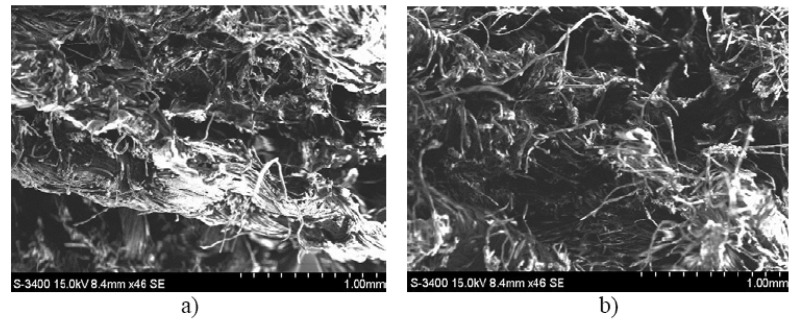
Image of the breaking area for a representative sample of the set of composite materials with matrix of hybrid resin of type 2 and reinforcing material of the first type of flax fabric (**a**), respectively second type of flax fabric (**b**).

**Figure 14 polymers-12-01649-f014:**
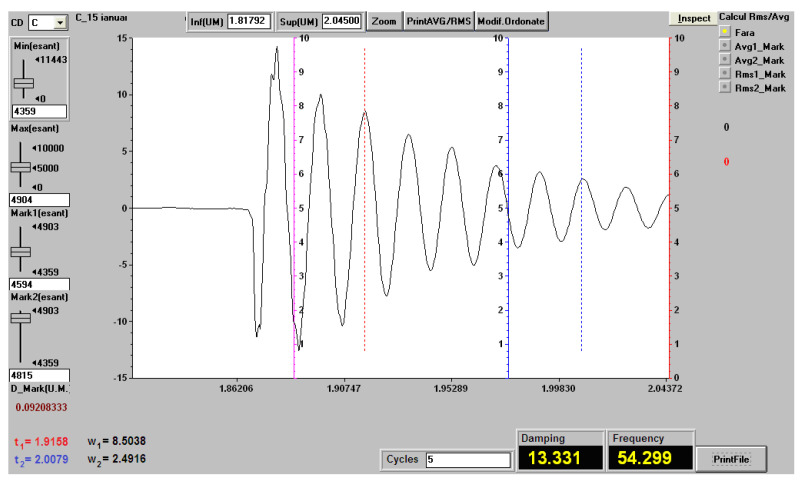
Registration of vibrations (natural frequency and damping factor) at a sample of hybrid resin of type 2, reinforced with first type of flax fabric, for a free length of 200 mm.

**Figure 15 polymers-12-01649-f015:**
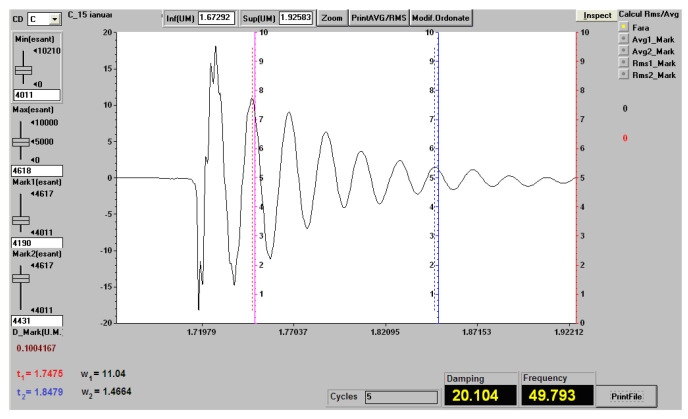
Recording of vibrations (natural frequency and damping factor) in a sample of hybrid resin of type 2, reinforced with the second type of flax fabric, for a free length of 200 mm.

**Table 1 polymers-12-01649-t001:** The constituents of the manufactured composite materials.

Hybrid Resin Type	Dammar Volume Ratio	Fabric Type	Number of Layers	Mass Ratio of Hybrid Resin
1	50%	80% flax and 20% cotton	12	0.52
		60% flax and 40% cotton	20	0.52
2	60%	80% flax and 20% cotton	12	0.54
		60% flax and 40% cotton	20	0.51
3	70%	80% flax and 20% cotton	12	0.50
		60% flax and 40% cotton	20	0.52

**Table 2 polymers-12-01649-t002:** The mean value with the standard deviation for elasticity modulus, tensile strength and elongation at break for samples of the 3 types of hybrid resin.

Type of	Elasticity Modulus	Tensile Strength	Elongation at Break
Hybrid Resin	*E* [N/mm${}^{2}$]	$R_{m}$ [MPa]	*A* [%]
1	3077 (± 92)	25.5 (± 1.5)	1.4 (± 0.08)
2	2444 (± 70)	19 (± 1)	1.96 (± 0.07)
3	1798 (± 45)	13 (± 1)	2.95 (± 0.1)

**Table 3 polymers-12-01649-t003:** 10 volatile compounds identified in the chemical structure of hybrid resin of type 2.

Current	Spectrum	Aldrich/Smiths Detection ATR
Number		Special Libraries
1	0.7628	Vetiver (Natural Essential Oil)
2	0.7563	Geranium algerie; Essence
3	0.7557	Geranium bourbon (Natural Essential Oil)
4	0.7407	Patchouli (Natural Essential Oil)
5	0.7333	Dammar Gum
6	0.7270	Vetiver java (Natural Essential Oil)
7	0.7002	Bergamot (Natural Essential Oil)
8	0.6988	Peppermint oil
9	0.6917	Dinonylnaphthalenesulfonic acid (in kerosene)
10	0.6900	Lavandin (Natural Essential Oil)

**Table 4 polymers-12-01649-t004:** Chemical structure of a specimen of hybrid resin sample of type 2, taken from sample 2.x.

Element	Element	Element	Atomic	Weight	Atomic
Number	Symbol	Name	Weight	Conc.	Conc.
				[%]	[%]
6	C	Carbon	12	83.8	87.51
8	O	Oxygen	16	15.7	12.29
17	Cl	Chlorine	35.5	0.3	0.11
11	Na	Sodium	23	0.1	0.06
20	Ca	Calcium	40	0.1	0.03

**Table 5 polymers-12-01649-t005:** The mean value with the standard deviation for elasticity modulus, tensile strength and elongation at break for composites with matrix of hybrid resin of type 1, 2 and 3, reinforced with the first type of flax fabric.

Type of Hybrid Resin	Thickness of Sample [mm]	Mass Ratio of Hybrid Resin	Elasticity Modulus *E* [N/mm${}^{2}$]	Tensile Strength $R_{m}$ [MPa]	Elongation at Break *A* [%]
1	6.3	0.52	5340 (± 50)	73.5 (± 1.5)	3.4 (± 0.1)
2	6.3	0.54	5180 (± 40)	70.5 (± 1.5)	3.28 (± 0.08)
3	6.2	0.50	5005 (± 55)	67 (± 1)	3.2 (± 0.1)

**Table 6 polymers-12-01649-t006:** The mean value with the standard deviation for elasticity modulus, tensile strength and elongation at break for composites with matrix of hybrid resin of type 1, 2 and 3, reinforced with second type of flax fabric.

Type of	Thickness of	Mass Ratio	Elasticity	Tensile Strength	Elongation
Hybrid Resin	Sample	of Hybrid Resin	Modulus		at Break
	[mm]		*E* [N/mm${}^{2}$]	$R_{m}$ [MPa]	*A* [%]
1	6.2	0.52	4890 (± 20)	69 (± 1)	2.82 (± 0.08)
2	6.1	0.51	4790 (± 50)	65.5 (± 1.5)	2.70 (± 0.05)
3	6.1	0.52	4720 (± 40)	62 (± 1)	2.58 (± 0.08)

**Table 7 polymers-12-01649-t007:** Behavior under vibrations of samples from composites with matrix of hybrid resin of type 1, 2 and 3, reinforced with the first type of flax fabric.

Free	Composite Material with		Composite Material with		Composite Material with	
Length	Hybrid Resin of Type 1		Hybrid Resin of Type 2		Hybrid Resin of Type 3	
	Frequency	Damping	Frequency	Damping	Frequency	Damping
[mm]	ν [Hz]	μ [s${}^{-1}$]	ν [Hz]	μ [s${}^{-1}$]	ν [Hz]	μ [s${}^{-1}$]
150	95.1	29.9	90.2	31.3	87.2	33.1
175	67.1	24.3	65.3	24.9	62.8	26.5
200	51.2	18.9	49.8	20.1	48.1	21.0
225	39.2	14.2	38.8	15.3	37.3	16.5
250	33.1	12.1	31.3	12.8	29.8	13.7

**Table 8 polymers-12-01649-t008:** The behavior under vibrations of samples from composites with matrix of hybrid resin of type 1, 2 and 3, reinforced with the second type of flax fabric.

Free	Composite Material with		Composite Material with		Composite Material with	
Length	Hybrid Resin of Type 1		Hybrid Resin of Type 2		Hybrid Resin of Type 3	
	Frequency	Damping	Frequency	Damping	Frequency	Damping
[mm]	ν [Hz]	μ [s${}^{-1}$]	ν [Hz]	μ [s${}^{-1}$]	ν [Hz]	μ [s${}^{-1}$]
150	101.0	22.3	96.3	23.3	91.3	24.2
175	73.4	17.7	70.6	18.1	65.9	18.9
200	56.2	13.2	54.3	13.3	51.5	14.4
225	43.4	10.0	42.1	10.6	40.6	11.4
250	35.3	8.8	34.1	9.1	31.9	9.4

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
