# Peer review of "A Study of the Mechanical Properties of Composite Materials with a Dammar-Based Hybrid Matrix and Two Types of Flax Fabric Reinforcement"

_polymers, 2020, doi:10.3390/polym12081649_

Round 1
Reviewer 1 Report
The work “A Study of Some Mechanical Properties of Composite Materials with a Dammar-based Hybrid Matrix and Two Types of Flax Fabric Reinforcement” is centered on the scope of the journal, novel and suitable to publish. I have some remarks that the authors should add or change, in particular I suggest to reorganize some results to improve the clarity.
Title: I suggest to change “some” in “the” or “tensile”
Pag 2 line 33 when the authors write dense do they mean viscous or are they talking about density?
Pag 2 line 61 I suggest to add at least one refs. about alkaline treatment, which can be considered one of the most common treatments on natural fibers. e.g.:
Reinforcing mechanisms of natural fibers in green composites: Role of fibers morphology in a PLA/hemp model system. Mazzanti V., Pariante R., Bonanno A., Ruiz de Ballesteros O., Mollica, Filippone G. Composites Science and Technology 180, 2019, pp. 51-59
In the materials section, for all the materials used, the manufacturer and trade name need to be add to the text
Pag 3 line 121 I suggest adding for clarity a table with all the materials done and the constituents of which they are made
Pag 4 line 131 I recommend to describe better the steps to obtain the cast plates? For example if pressure or temperature has been applied
Pag 5 line 162 the speed of the test and the number of sample used need to be add in the text
Pag 5 line 165 Replace “elements” with mechanical properties
Pag 6 The average value of the tensile test need to be listed only in the table 3 and not in the text (line 187- 189- 191) with their standard deviation (not the range)
Fig. 4- 6 I suggest to the authors to use only a plot with the three tensile characteristic curves to facilitate the visualization of the differences between the three resins
Fig.7 this figure is a bit "crowded", perhaps the authors could underline only the most significant peaks
Fig. 8 the axes of this figure are almost illegible
Fig.9-11 I suggest to the authors to use only a plot with the three tensile characteristic curves to facilitate the visualization of the differences between the three composites
Pag 9 The average value of the tensile test need to be listed only in the table 4 and not in the text (line 225-227-229) with their standard deviation (not the range or limits)
Fig.12-14 I suggest to the authors to use only a plot with the three tensile characteristic curves to facilitate the visualization of the differences between the three composites
Pag 11 The average value of the tensile test need to be listed only in the table 5 and not in the text (line 236-238-240) with their standard deviation (not the range or limits)
Fig.15 SEM analyzes become much clearer if a gold coating is sprayed on the sample. Did you have performed this type of treatment?
Pag 13 288-290 The tensile test results need to be express under the form of average not like a range
In the discussion the authors need to discuss deeply the effect of the fibers comparing the result of the composites samples with the neat ones
SEM analysis are not discussed in the discussion section
The authors must divide the discussions from the conclusions section. The conclusions must briefly summarize the main and most significant results of the paper.
Author Response
Dear Reviewer 1,
Based on your comments, we have made the following changes to the structure of the paper with
Title: “A STUDY OF THE MECHANICAL PROPERTIES OF COMPOSITE MATERIALS WITH A DAMMAR-BASED HYBRID MATRIX AND TWO TYPES OF FLAX FABRIC REINFORCEMENT”
Authors: Dumitru Bolcu, Marius Marinel Stănescu*
Point 1: Title: I suggest to change “some” in “the” or “tensile”;
Response 1: We changed "some" to "the".
Point 2: Pag 2 line 33 when the authors write dense do they mean viscous or are they talking about density?
Response 2: We changed “dense” to “viscous”.
Point 3: Pag 2 line 61 I suggest to add at least one refs. about alkaline treatment, which can be considered one of the most common treatments on natural fibers. e.g.: Reinforcing mechanisms of natural fibers in green composites: Role of fibers morphology in a PLA/hemp model system. Mazzanti V., Pariante R., Bonanno A., Ruiz de Ballesteros O., Mollica, Filippone G. Composites Science and Technology 180, 2019, pp. 51-59;
Response 3: In the introduction to the paper we presented the advantages of alkaline fiber treatment and made the bibliographic reference.
Point 4: In the materials section, for all the materials used, the manufacturer and trade name need to be add to the text;
Response 4: We introduced in the text of the paper as bibliographic references the webpage of the Dammar resin distributor (www.foitadeaur.ro/rasini.htm) and that of the manufacturer of the two types of flax fabric (http://dinatex.3x.ro).
Point 5: Pag 3 line 121 I suggest adding for clarity a table with all the materials done and the constituents of which they are made;
Response 5: At the end of subsection 2.1 we have introduced table 1 with the required data about the materials used.
Point 6: Pag 4 line 131 I recommend to describe better the steps to obtain the cast plates? For example if pressure or temperature has been applied;
Response 6: At the beginning of subsection 2.1 we made the specification regarding the temperature of the environment in which we made the casting (21-230C). In addition, we also mentioned that when pouring the composite plates we applied a uniform pressure of 27000 N / m2.
Point 7:Pag 5 line 162 the speed of the test and the number of sample used need to be add in the text;
Response 7: We specified the speed used for traction (2.5 mm / min) and the number of samples subjected to the tensile test.
Point 8: Pag 5 line 165 Replace “elements” with mechanical properties;
Response 8: We replaced "elements" with "mechanical properties".
Point 9: Pag 6 The average value of the tensile test need to be listed only in the table 3 and not in the text (line 187- 189- 191) with their standard deviation (not the range);
Response 9: We have removed from the text, according to figure 4, the values ​​of the breaking stress, the breaking elongation and the modulus of elasticity. We have entered these data in Table 2.
Point 10: Fig. 4- 6 I suggest to the authors to use only a plot with the three tensile characteristic curves to facilitate the visualization of the differences between the three resins;
Response 10: We put the three characteristic curves from hybrid resins in a single graph. Figures 4-6 became figure 4 after renumbering.
Point 11: Fig.7 this figure is a bit "crowded", perhaps the authors could underline only the most significant peaks;
Response 11: The diagram in figure 7 (which became after renumbering figure 5) is automatically drawn by the software of the FTIR analysis system (for simplification we have removed the list of peaks).
Point 12: Fig. 8 the axes of this figure are almost illegible;
Response 12: We tried to improve the quality of the diagram in figure 8 (which became after renumbering figure 6), by choosing appropriate colors. The diagram is automatically drawn by the EDS analysis software.
Point 13: Fig.9-11 I suggest to the authors to use only a plot with the three tensile characteristic curves to facilitate the visualization of the differences between the three composites;
Response 13: The characteristic curves of composite materials made with the first type of flax fabric (figures 9-11 became after renumbering figures 7-9) are presented separately because in the case of their representation in a single figure we found that there are overlaps of the characteristic curves and not a clear distinction can be made between them.
Point 14: Pag 9 The average value of the tensile test need to be listed only in the table 4 and not in the text (line 225-227-229) with their standard deviation (not the range or limits);
Response 14: We have removed from the text, according to figures 7-9, the values ​​of the breaking stress, the breaking elongation and the modulus of elasticity. We have entered these data in Table 5.
Point 15: Fig.12-14 I suggest to the authors to use only a plot with the three tensile characteristic curves to facilitate the visualization of the differences between the three composites;
Response 15: The characteristic curves of composite materials made with the second type of flax fabric (figures 12-14 became after renumbering figures 10-12) are presented separately because in the case of their representation in a single figure we found that there are overlaps of the curves characteristics and their clear distinction can no longer be made.
Point 16: Pag 11 The average value of the tensile test need to be listed only in the table 5 and not in the text (line 236-238-240) with their standard deviation (not the range or limits);
Response 16: We eliminated the text, after the numbers 10-12, capitalize on the stresses related to rupture, elongations at break and the modulus of elasticity. We have entered the date given in Table 6.
Point 17: Fig.15 SEM analyzes become much clearer if a gold coating is sprayed on the sample. Did you have performed this type of treatment?
Response 17: We did not perform this type of gold dust treatment on SEM analysis (for financial reasons).
Point 18: Pag 13 288-290 The tensile test results need to be express under the form of average not like a range;
Response 18: We expressed the results of the traction test below average.
Point 19: In the discussion the authors need to discuss deeply the effect of the fibers comparing the result of the composites samples with the neat ones;
Response 19: We highlighted the effect of flax fibers in the mechanical behavior of the studied composites. The properties of flax fibers are influenced not only by natural factors (climatic conditions, soil, seed varieties, etc.), but also by plant processing conditions. We can find, based on bibliographic data, that the mechanical properties of flax fibers (see [Mohanty 2000] and [Mohanty 2005] in the bibliography) are far superior to the mechanical properties of the hybrid resins studied by us. And in the case of studied composite materials, we found that the values ​​of modulus of elasticity and those of tensile strength are much higher than those of hybrid resins used as matrices. Therefore, we can conclude that the two types of flax fabrics have a decisive influence in establishing their mechanical behavior. In addition, the elongation at break in composite materials reinforced with the two types of flax fabric is close to the elongation at break of flax fibers which is 2.7-3.2% (see [Mohanty 2000] and [Mohanty 2005]). Therefore, the breaking of the test pieces from the studied composite materials took place at the time of breaking the fibers.
Point 20: SEM analysis are not discussed in the discussion section;
Response 20: In the paragraph "Discussions" some comments are made in connection with figure 15 (which became figure 13 after renumbering). The behavior of composite materials is also confirmed by the images obtained based on SEM analysis, in which is observed that the fibers, in the breaking area, are detached from the matrix. In addition, a higher density of fibers is observed in the case of the first type of flax fabric, which explains the superior properties of composites reinforced with this type of fabric.
Point 21: The authors must divide the discussions from the conclusions section. The conclusions must briefly summarize the main and most significant results of the paper.
Response 21: We separated the "Discussions" and "Conclusions" sections.
We mention that the responses addressed to reviewer 1 are colored in the text of the paper in blue, and the answers addressed to both reviewers are colored in brown.
Thanks for the views expressed on the basis of which we have made the changes that have contributed to increasing the scientific level of the paper.
Authors
Please see the attachment.

Reviewer 2 Report
What are the typical properties of Dammar resin?
Please provide the parameters of tensile testing.
Quality of the presentation should be higher. For example Figures 4-6 should be compiled into one figure. Same for 9-11 and 12-14. Quality of the Figure 7 should be improved.
Regarding tensile tests, Authors could calculate and present the values of resins and composites toughness.
Why higher shares of Dammar resin are resulting in higher ductility? Please provide the explanation, possibly based on the chemical structure of both resins.
How Authors determined the volatile organic compounds in resin?
Why only the results for hybrid resin 2 are presented?
Results for composite materials should be discussed and compared with each other.
Figure 15 - some explanation and discussion should be provided.
Similar for the results of vibration tests. Except the presentation of the results, they should be discussed.
Author Response
Dear Reviewer 2,
Based on your comments, we have made the following changes to the structure of the paper with
Title: “A STUDY OF THE MECHANICAL PROPERTIES OF COMPOSITE MATERIALS WITH A DAMMAR-BASED HYBRID MATRIX AND TWO TYPES OF FLAX FABRIC REINFORCEMENT”
Authors: Dumitru Bolcu, Marius Marinel Stănescu*
Point 1: What are the typical properties of Dammar resin?
Response 1: There are no significant results regarding the mechanical properties of Dammar natural resin. This is due to the shape of the granules in which the Dammar resin is found, which does not allow the making of samples to be subjected to the tensile test. In the literature there is research on the composition and chemical behavior of natural resin Dammar ([Hidayat, Ukiya] in the bibliography). There are few studies on the mechanical properties of Dammar natural resin, but these refer to mixtures (for obtaining films, modified silicone, etc. [Pethe] and [Zakaria 2012-9, Zakaria-2012-42] from the bibliography).
Point 2: Please provide the parameters of tensile testing;
Response 2: We specified that the speed used for traction was 2.5 mm / min.
Point 3: Quality of the presentation should be higher. For example Figures 4-6 should be compiled into one figure. Same for 9-11 and 12-14. Quality of the Figure 7 should be improved;
Response 3: We put the three characteristic curves from hybrid resins in a single graph. Figures 4-6 became after renumbering figure 4. The characteristic curves of the composite materials made with the two types of flax fabric (figures 9-11 and figures 12-14, which became after renumbering figures 7-9 and figures 10-12) are presented separately because in the case of their representation in a single figure we found that there are overlaps of the characteristic curves and it is no longer possible to make a clear distinction between them.
The diagram in figure 7 (which became after renumbering figure 5) is automatically drawn by the software of the FTIR analysis system (for simplification we have removed the list of peaks).
Point 4: Regarding tensile tests, Authors could calculate and present the values of resins and composites toughness;
Response 4: Based on the tensile stress, the tensile strength, the modulus of elasticity and the breaking elongation at break are determined for all the samples made.
Point 5: Why higher shares of Dammar resin are resulting in higher ductility? Please provide the explanation, possibly based on the chemical structure of both resins;
Response 5: We explained the changes that occur in the behavior of hybrid resins as the volume ratio of Dammar increases. More precisely, resins even in solid state have a rheological behavior characterized by viscosity. Due to external stresses, a low viscosity leads to significant deformations. Dammar resin diluted with turpentine remains liquid, so it has a low viscosity. Solidification occurs by mixing with synthetic resin. Therefore, the viscosity of the hybrid resin obtained decreases with increasing volume ratio of Dammar. Therefore, at the same external load, the deformations of the hybrid resin by 70% Dammar are higher than the deformations of the hybrid resin by 60%, respectively 50% volume ratio of Dammar.
Point 6: How Authors determined the volatile organic compounds in resin?
Response 6: The FTIR analysis equipment is certified with Aldrich / Smiths ATR detection spectral libraries that can automatically identify a maximum of 10 volatile organic compounds in the structure of the analyzed resin.
Point 7: Why only the results for hybrid resin 2 are presented?
Response 7: In-depth studies on the composition and chemical properties of Dammar-based hybrid resin (with various volume ratio) were performed in [Neda] in the literature. We chose to study the type 2 hybrid resin (in terms of structure), because it falls within the average chemical properties of the three types of hybrid resin.
Point 8: Results for composite materials should be discussed and compared with each other;
Response 8: By comparison, composites made with the two types of flax fabrics have similar behaviors. However, composites reinforced with the first type of fabric have a breaking strength with 4.5-5 MPa higher than composites reinforced with the second type of fabric. And for the modulus of elasticity the conclusion is similar. Composites reinforced with the first type of fabric have a modulus of elasticity with 285-450 MPa higher than composites reinforced with the second type of fabric. These differences arise from variations in the properties and distribution of flax fibers in the fabric, respectively.
Point 9: Figure 15 - some explanation and discussion should be provided;
Response 9: In the paragraph "Discussions" we made some comments about figure 15 (which after renumbering became figure 13). The behavior of composite materials is also confirmed by the images obtained using SEM analysis, which shows that the fibers, in the breaking area, are detached from the matrix. In addition, a higher density of fibers is observed in the case of the first type of flax fabric, which explains the superior properties of composites reinforced with this type of fabric.
Point 10: Similar for the results of vibration tests. Except the presentation of the results, they should be discussed;
Response 10: The natural frequencies of the bars depend on not only the dimensions (thickness, length), but also on the material properties (density and modulus of elasticity). Because the test samples studied had similar dimensions, the differences between the measured frequencies are due to the differences between the modulus of elasticity of the composite materials from which the samples are made. We found that there is a correspondence between the measured frequencies and the modulus of elasticity determined at the tensile test, more precisely the increase of the modulus of elasticity leads to the increase of the vibration frequency. The damping factor as a whole characterizes the vibration damping capacity for a test piece. Comparison of the results in Tables 7 and 8 shows that the samples reinforced with the second type of fabric dampen the vibrations better. We also found that the vibration damping capacity increases with increasing volume ratio of Dammar in the hybrid resin used as a matrix.
We mention that the responses addressed to reviewer 2 are colored in the text of the paper in green, and the answers addressed to both reviewers are colored in brown.
Thanks for the views expressed on the basis of which we have made the changes that have contributed to increasing the scientific level of the paper.
Authors
Please see the attachment.

Round 2
Reviewer 2 Report
Ok for me now